# Analyzing the evolutionary trajectory of technological themes based on the BERTopic model: A case study in the field of artificial intelligence

Xinle Zheng[1☺], Qing Wu[1,2☺], Xingyu Luo[3☺], Kun Lv[1,2☺]*

1 Business School, Ningbo University, Ningbo, China, 2 Merchants' Guild Economics and Cultural Intelligent Computing Laboratory, Ningbo University, Ningbo, China, 3 School of Information Management, Central China Normal University, Wuhan, China

☺ These authors contributed equally to this work.
* lvkun@nbu.edu.cn

## Abstract

As the wave of technological innovation propels national development into the future, technological advancement has emerged as a crucial pillar for enhancing international competitiveness. Unraveling the evolutionary trajectory of technologies and their associated themes provides a solid theoretical foundation for strategic decision-making in fostering future industrial technological upgrades, thereby aiding in seizing the initiative in technological innovation. This study, adopts a multi-source data perspective, employing the life cycle theory to delineate temporal windows. We use the BERTopic model to extract technological themes and construct a co-occurrence network of theme keywords. Three network centrality indices are computed to filter key theme terms, and the Word2Vec model is leveraged to calculate cosine similarities. Ultimately, we map out the evolutionary pathway of technological themes using Sankey diagrams. Taking the field of artificial intelligence as an example, the study found that the proposed method could effectively identify 48 technical theme keywords and analyze the technological evolution paths of these keywords in areas such as scenario applications, network services, human-computer interaction, intelligent detection, and natural language processing. Furthermore, all evaluation metrics of the model outperformed those of comparable topic models. The rationality of the empirical results was validated through examination against national policies and market application scenarios.

## Introduction

Strengthening the "From 0 to 1" Fundamental Research Initiative underscores that, amidst the vigorous rise of the scientific and technological revolution and industrial transformation, the advancement of science and technology has emerged as

**Data availability statement:** All relevant data are within the manuscript and its Supporting information files.

**Funding:** This research was supported by: The 2024 Ningbo Soft Science Research Program: Research on the Resource Carrying and Utilization of Ningbo Science and Technology Celebrities - Taking the Revitalization and Development Path of Ningbo's "Hometown of Academicians" as an Example (2024R044); The Zhejiang Soft Science Research Plan Project "Boosting the Development of Zhejiang's Low-Altitude Economy: Current Situation Analysis, Application Scenario Exploration, and Countermeasure Research" (2025C35059); The Merchants' Guild Economics and Cultural Intelligent Computing Laboratory of Ningbo University has set up its own project in 2025: Identification and Value Capture of First Economic Effects under Business Group Network: Attention Economic Analysis and Causal Inference Based on Large Model ( MGECICL2025T11 ).

**Competing interests:** The authors have declared that no competing interests exist.

a pivotal focal point of international competition and a crucial fulcrum for national strategic deployment. Technological innovation plays a pivotal role in major special programs, driving societal transformation [1]. In response to national policy calls, an increasing volume of resources is being invested in theoretical research and practical applications of technological innovation, resulting in a continuously growing and intricately diverse array of scientific achievements across various disciplines. Concurrently, with the unabated progression of the wave wherein technological innovation spearheads industrial innovation, the profound integration of new paradigms and elements within the industrial revolution and scientific and technological innovation has injected robust impetus into economic development and the enhancement of advanced productivity [2].

Furthermore, as an integral component of national strategy, artificial intelligence spans multiple industrial sectors including finance, healthcare, manufacturing, and electronic information, serving as a cornerstone in propelling the nation's digital economy and fostering digital industrial clusters, thereby exerting profound influence in future international competitions. Consequently, the rapid and precise identification of technological development priorities, recognition of technological themes, and elucidation of their evolutionary trajectories through scientific methodologies constitute effective approaches to achieving deep integration between scientific and technological innovation and industrial innovation, laying a solid foundation for future technological strategic planning [3]. However, the exponential growth in the quantity of scientific research outcomes, driven by interdisciplinary convergence and market diversification, poses heightened demands for technological opportunity identification. The construction of a comprehensive, multi-tiered technological theme research model has thus emerged as a focal point of current research endeavors [4].

Against this backdrop, this study, adopting a multi-source data perspective, proposes a methodology that amalgamates life cycle theory with the BERTopic model to swiftly and accurately discern technological development priorities, identify technological themes, and clarify their evolutionary trajectories. Compared to single-source data, our approach offers a broader spectrum of technological information, facilitating in-depth exploration of cross-disciplinary knowledge linkages both within and external to technologies, thereby forging a more holistic knowledge network. In summary, this study pioneers the integration of multi-source data information for application in technological theme research. By constructing a co-occurrence network of thematic terms, calculating network centrality metrics for thematic term screening, and subsequently employing the Word2Vec model to compute cosine similarities for Sankey diagram visualization of technological theme evolutionary paths, these findings not only establish a theoretical foundation for strategic deployment in industrial technological innovation but also provide intelligence support for relevant technical personnel and regulatory bodies.

## Literature review

### Research on multi-source data

In the context of data-driven decision-making, the effective integration and utilization of multi-source data are prerequisites for the sustainable development of China's

industrial competitive intelligence smart service system. Multi-source data, derived from diverse channels, serve as carriers of various knowledge flows. Essentially, these data exhibit differences in attributes such as data structure and content, leading to variations in the technical theme knowledge they contain. While the majority of scholars currently rely on single data sources for technology opportunity identification research, a subset of researchers integrates multi-source data, including patents, papers, news, reviews, and funding, for correlation studies. Xu Xueming combined datasets from three types of sources—literature, patents, and news—to compress technical themes and screen out key technical themes aligned with technological development trajectories from both horizontal and vertical dimensions [5]. Liu Xiwen explored data from four sources: papers, patents, startup company data, and public opinion, to mine and generate a list of candidate technologies and assess their disruptive potential [6]. Liu Yufei conducted time slicing based on ordered clustering of multi-source samples, calculated similarities between clusters at different levels, and derived the evolutionary paths of technical themes [7]. In summary, compared to single data sources, multi-source data provide richer knowledge associations and finer-grained analysis in the identification of technical themes and opportunities. At the level of technological evolution, multi-source data enable more precise mining of potential technologies and exhibit superior generalization capabilities.

## Research on technological themes

Technical themes, as data sets that most intuitively reflect technical content and directions, are currently hot topics of research for many scholars. Early studies primarily focused on analyzing structured characteristic data, such as keywords, classification codes, and citation relationships within scientific and technological literature. These studies employed methods such as the Delphi technique, keyword co-occurrence analysis, and citation path analysis to identify associations and co-occurrence features among technologies and their themes. Momeni A et al. [8] further refined this approach by utilizing patent development pathways, k-core analysis, and topic modeling (ClusTop) to identify core technologies. Building upon these foundations, some scholars combined these methods in their research. Faraji O et al. [9] and Dai et al. [10] used co-word analysis, social network (SN) analysis, and multidimensional scaling (MDS) to reveal gaps in the literature and propose suggestions for future technological research. As the number of scientific research outcomes continued to increase, bibliometric indicators such as citation frequency, the H-index [11], and annual publication volume emerged as methods for identifying important technologies and their themes. Bortoluzzi M et al. [12] and Yang et al. [13] considered key performance indicators, similarity, and coherence as crucial metrics for evaluating technical themes. By calculating the number of publications and the strength of thematic associations over specific time periods, they identified trends in the development of technical themes, which had significant implications for corporate technology strategies. With the innovative development of natural language technologies, unstructured text data in scientific and technological literature has played an increasingly important role in the study of technical themes. Scholars have gradually expanded their analysis from external characteristics of scientific and technological literature to deeper semantic aspects. LDA (Latent Dirichlet Allocation), as one of the classic text topic models, is simple, easy to operate, and highly interpretable. It is currently used by many scholars in related research on technical themes, often combined with methods such as text clustering and social network analysis. Sun et al. [14] used LDA as the basis for topic clustering, combined with an integrated algorithm of non-negative matrix factorization and the mean-shift algorithm, resulting in higher accuracy and stability in technical topic clustering. Hu et al. [15] combined the LDA topic model with the ARIMA model to predict the prevalence and development trends of technical themes. Moreover, the complex and diverse content of scientific and technological literature has driven improvements and integrations of the LDA model. The TD-LDA model based on probabilistic graphs and intensity calculation models [3], the technical indicator system built upon LDA-DEMATEL-ISM [16], and the importance index defined based on ICA-LDA and weighted average techniques [17] provide a solid theoretical foundation for related research on technical themes. These advancements not only deepen our understanding of technical themes but also offer valuable insights for guiding future technological research and development.

## Research on the evolution path of technological themes

Building upon the foundation of previous research on technical themes, the evolution of these themes is typically preceded by a temporal segmentation, where the associations among technical themes across different time periods are examined to uncover the development trajectory and evolutionary patterns within specific technological domains. This represents the continuous transformation of technological innovation. The visualization of the evolution path of technical themes, also known as technical theme trajectories or technological trajectories, was conceptualized by Dosi based on the notion of "technological paradigms" [18].In early studies, co-citation analysis was employed to reveal the flow of knowledge within technological fields. Zhang Ben et al. utilized main path analysis to extract the evolution path of citations as clusters within a co-citation network, from which nodes of the main path network were obtained [19]. On this basis, scholars have classified the evolution of technical themes into six types: merging, splitting, expanding, contracting, emerging, and disappearing [20–22]. These classifications are visually represented through the association strength, thematic similarity, or co-occurrence frequency relationships of technical themes across different time sequences. Cobo proposed a method to quantify and visualize the evolution of technical themes by linking technical themes across different time periods through co-word analysis, using density and centrality parameters to characterize their positions and the h-index to measure the impact of themes and their domains [23] Kwak et al. detected anomalies in CAN bus systems using cosine similarity, providing a safety guarantee for the innovative evolution of technical themes [24].Furthermore, with the support of bibliometric analysis, some scholars have conducted systematic reviews of thin-film solar cells (TFSC), employing topic modeling and technical theme citation networks to extract the research landscape and trends of technical themes [25]. Zhang et al., based on the Octree-based Topic Model (OCTM), explored the types of topic evolution and identified paths by combining the co-occurrence coefficient between topics within the same time window and the similarity between topics in adjacent time windows [26]. Cheng Xiufeng et al. applied fused word embedding technology to a keyword network based on "citation co-occurrence" relationships, resulting in more sustained and logical evolutionary paths [27].

## Literature review and commentary

Through the aforementioned literature review, this study reveals that current research on technological theme evolution typically concentrates on studies utilizing single-source data, employing methodologies such as the Delphi method, bibliometrics, text analysis, and social network analysis for technological theme identification, and leveraging structured data to dissect the external characteristics of technologies. In terms of semantic analysis, scholars predominantly utilize the Latent Dirichlet Allocation (LDA) model or text clustering methods to identify technological themes and their associated thematic terms, focusing solely on semantic associations between individual words or phrases, yet deeper exploration is warranted for semantic analysis within contextual frameworks. In contrast, the BERTopic thematic model not only considers the co-occurrence frequency of words but also captures semantic variations of words across different contexts through deep learning techniques, rendering the identification of technological themes more precise and nuanced. Concurrently, the Word2Vec model demonstrates its superiority in excavating the dynamic knowledge flow processes among themes across various timelines. Specifically, by mapping words to a high-dimensional vector space, it generates a context-based distributed representation (i.e., word embeddings) for each word. These word embeddings not only capture semantic relationships between words but also unveil similarities and associations among them through mathematical operations. Therefore, from the perspective of multi-source data analysis, this study delves into technological themes by employing the technology life cycle theory to delineate temporal phases, utilizing advanced semantic models to explore technological themes from a contextual association standpoint, leveraging their vectorial characteristics for similarity associations, and further excavating the dynamic knowledge flow processes among themes across different timelines. This approach facilitates a fine-grained analysis of the evolutionary processes in related domains, providing a robust theoretical foundation for future strategic decision-making. In summary, the methodology adopted in this study exhibits distinct advantages in terms

of accuracy, semantic comprehension, and dynamic knowledge flow analysis, representing a novel and effective exploration in the realm of technological theme evolution research.

## Research design

### Research framework

This study proposes a method for analyzing the evolution path of technological themes based on multi-source data from the perspective of semantic analysis. It divides the time stages of technological theme development through the technology life cycle theory, extracts abstract data from relevant scientific and technological literature for Bertopic theme modeling, screens technological theme words based on the calculation of three network centrality indices, employs the Word2Vec model for technological theme vectorization, and calculates their cosine similarities. The evolution path of technological themes is visualized through Sankey diagrams. The specific process consists of the following five steps: Firstly, through literature review and paper references, literature search formulas are summarized to obtain paper, patent, and grant data from CNKI, One Patent, and Wanfang Data's SCI database, respectively, while screening out abstracts lacking and irrelevant literature. Secondly, the technology life cycle theory is applied to divide the time stages of the corresponding literature. Thirdly, technological themes are identified through Bertopic theme modeling. Fourthly, a co-occurrence network of theme words is constructed, and network centrality indices such as degree centrality, closeness centrality, and betweenness centrality are calculated to screen theme words. Theme word associations are then established using the Word2Vec model and cosine similarity. Fifthly, based on the similarity of theme words across different time sequences, a Sankey diagram is constructed to analyze the evolution path of technological themes and related knowledge flows. The research framework is shown in Fig 1.

### Data sources and data acquisition

Based on the research framework, the foremost priority is to select appropriate data to ensure the accuracy and comprehensiveness of the study.As the largest academic paper database in China, CNKI includes over 95% of officially published Chinese academic resources, covering knowledge content across multiple disciplines.Timely data updates and highly current literature, aid in fully understanding the research status and knowledge flows within specific fields. One Patent

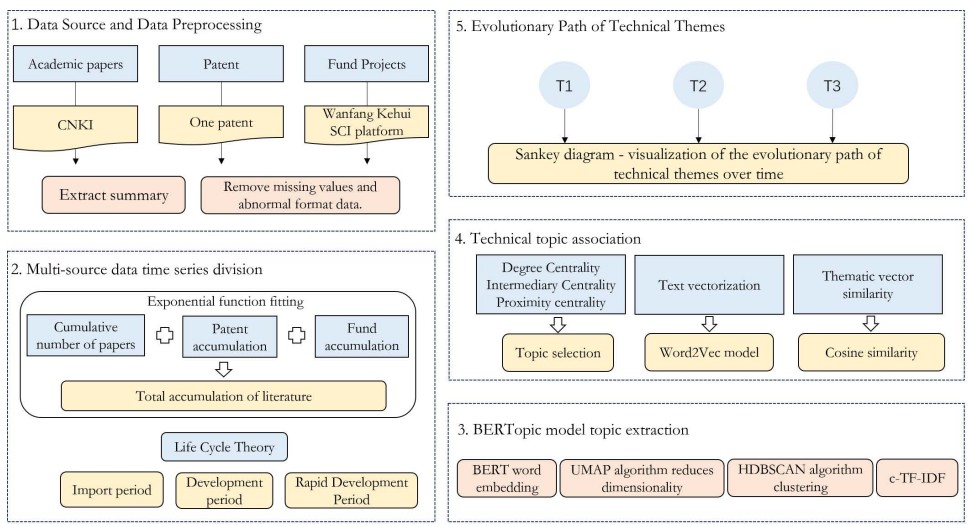

**Fig 1. Research Framework.**

currently includes over 179 million global patents spanning more than 150 countries. Combining domestic and international technological advantages, it is dedicated to providing efficient and precise information data retrieval for research institutions, universities, enterprises, and other organizations, containing rich and comprehensive data to meet various patent analysis needs. Wanfang Data's Sci-Fund platform dynamically tracks more than 7 million scientific research projects from over 200 funding agencies, establishing a multidisciplinary, comprehensive, and standardized scientific research project database to support scientific research innovation services. Therefore, This study utilizes the CNKI database, One Patent database, and Wanfang Data's Sci-Fund platform as data sources. Through data inquiries and considering the timeliness of the literature, search formulas for papers, patents, and fund projects are constructed to collect relevant literature data. After saving the data to a local database, documents that do not meet the research requirements due to incomplete abstracts or entirely unrelated content are manually screened out.This process not only safeguards the quality of the data but also lays a robust foundation for subsequent multi-source data analysis.

## Temporal division of multi-source data

The life cycle theory originates from natural ecosystems and is used to explain the full-cycle process of things from their inception, growth, maturity, and eventual decline. Currently, this theory is widely applied in various fields such as technology, economics, and society. The corresponding technology life cycle theory expresses the evolution of a technology over time. A technology undergoes four stages in its life cycle: introduction, growth, maturity, and decline, typically following an S-curve pattern [28,29], hence the derivation of the S-curve model for the technology life cycle. During the introduction stage, technology grows slowly, with a relatively small proportion of scientific research achievements, and the S-curve slope is relatively gradual. Entering the growth stage, the technology undergoes significant improvements and substantial performance enhancements, with the S-curve growth rate showing a marked upward trend. In the maturity stage, the technology boasts a vast amount of scientific research, the field gradually becomes saturated, and the growth rate slows down. Until the decline stage, the growth of technology in the field slows down again. Furthermore, some scholars have proposed that technological development can exhibit multiple S-curve progressions, experiencing growth-decline-renewal cycles. Considering actual domain scenarios, the current growth trends in the number of papers, patents, and fund projects clearly show rapid development, with relevant scholars finding exponential growth [30]. Therefore, This study uses an exponential function to fit the annual cumulative amounts of paper, patent, and fund data and employs certain indicators for measurement to test the fitting effect. The specific exponential function is shown in equation (1).

$$y = Ae^{Bx}$$

(1)

Here, $y$ represents the annual cumulative data corresponding to different data types; $x$ represents time. $A$ is a parameter, which is the intercept of the function with the y-axis when $x = 0$; $B$ is a parameter, reflecting the degree of change in $y$ as $x$ varies. The coefficient $R^2$ is used to judge the fitting effect of the model. The larger the $R^2$, the better the fitting effect of the model.

## Topic extraction using the BERTopic model

**BERTopic topic model.** The BERTopic topic model is a pre-trained topic model for text proposed by Grootendorst [31]. It can map large-scale documents into a topic space. Compared with traditional topic models such as LDA (Latent Dirichlet Allocation), LSA (Latent Semantic Analysis), and PLSA (Probabilistic Latent Semantic Analysis), the BERTopic model can deeply mine contextual semantic connections and demonstrates better performance in processing long-text corpus information. Relevant scholarly research and comparisons have found that the BERTopic topic model exhibits good recognition effects among scientific literature topic modeling methods [32–34] and has verified the applicability of this model in studies involving different text data sources such as patent topics [35], research paper topics [36], and

user demand topics [37]. This topic model employs the embedded BERT (Bidirectional Encoder Representations from Transformers) method to perform vectorized training on text data to obtain deep semantic vectors. It then uses the UMAP (Uniform Manifold Approximation and Projection) algorithm to perform dimensionality reduction analysis on the text vectors. The word vectors are clustered using the HDBSCAN (Hierarchical Density-Based Spatial Clustering) clustering method. Finally, c-TF-IDF (class-based Term Frequency-Inverse Document Frequency) is used to adjust the granularity of the topic clusters and extract topic information.

**Word embedding.** BERT, as an unsupervised training model, fully considers the semantic information of the context and excels at capturing textual semantic associations. Additionally, based on its multi-layer Transformer modules, it can process semantic data at different levels with strong generalization capabilities, while also demonstrating good performance in sentence-level vectorization.

**Dimensionality reduction using UMAP.** The UMAP (Uniform Manifold Approximation and Projection) dimensionality reduction algorithm models data using a fuzzy topological structure. It is a nonlinear dimensionality reduction technique. Compared to PCA, which may lose low-level structural information to improve computational speed, or T-SNE, which preserves data structural information well but slows down computational speed, the UMAP algorithm significantly improves data processing time while balancing the preservation of global data structure.

**HDBSCAN clustering algorithm.** The HDBSCAN algorithm, proposed by Campello et al. [38], essentially converts the DBSCAN algorithm into a hierarchical clustering algorithm by extending it through the extraction of flat clusters. Firstly, a spatial transformation is performed based on the density of the topic vectors, where the number of neighbor nodes within a smaller radius is taken as the density value. High-density regions are identified and separated from low-density regions. Secondly, a minimum spanning tree based on a weighted distance graph is constructed, with topic vectors as vertices and the edge weight between any two points representing topic similarity, also known as mutual reachability distance. By setting progressively changing thresholds, the topic hierarchy is separated. Then, a minimum cluster size parameter is set, and the hierarchy is traversed for splitting, achieving the effect of compressing the clustering tree. Finally, clustering extraction is performed. The advantage of the HDBSCAN algorithm is that it does not require manual setting of radius and neighborhood sample thresholds, and can automatically calculate the optimal cluster results. It effectively identifies abnormal data points and reduces noise, while providing insights into the distribution of different topics.

**C-TF-IDF.** TF-IDF, as a word weighting algorithm, is widely used in topic extraction research. The BERTopic model improves upon it by extending the c-TF-IDF algorithm. Unlike traditional TF-IDF, which identifies the importance of words between documents, c-TF-IDF treats all documents within the same cluster as a single document and calculates the importance of words within the cluster. This makes the extracted topics more representative while reducing computation time and improving overall efficiency, as shown in equation (2).

$$W_{x,c} = \| tf_{x,c} \| \times \log(1 + \frac{A}{f_x})$$

(2)

$tf_{x,c}$ represents the frequency of word $x$ in cluster $c$, $f_x$ represents the frequency of word $x$ across all clusters. $A$ represents the average number of words per cluster, $W_{x,c}$ represents the importance score of word $x$ in cluster $c$.

## Technical topic association

**Construction of co-occurrence network.** Co-occurrence networks represent a method for characterizing the relationships between elements by calculating the frequency with which two elements appear together within the same or similar contexts. In such networks, the elements serve as nodes, while the frequency of their co-occurrence relationships forms the edges. Meanwhile, in social network analysis, degree centrality, betweenness centrality, and closeness centrality are metrics employed to gauge the significance of nodes within a network. Degree centrality measures the extent of a node's connectivity within the network, i.e., the node's degree; the higher the degree of a node, the greater

its degree centrality, indicating that the node has more direct connections within the network. Betweenness centrality assesses the degree to which a node acts as an intermediary for information transmission within the network, often situated on critical paths that exert considerable influence over information flow. Closeness centrality evaluates the average distance between a node and other nodes in the network; the closer a node is to other nodes, the greater its closeness centrality, suggesting that the node can more readily communicate with other nodes, either directly or indirectly, within the network. In contrast, traditional research on technological theme identification often confines itself to the direct associations of individual words or phrases, overlooking their intricate semantic relationships in diverse contexts. By comparison, co-occurrence networks not only account for the frequency statistics of words but also reveal deeper interaction patterns among words through network nodes (thematic terms) and edges (co-occurrence relationships). This graph-theoretic approach can visually demonstrate the closeness and structural characteristics of thematic terms, facilitating the discovery of latent knowledge flow pathways and technological development trends. Furthermore, by calculating network centrality metrics, we can identify key thematic terms, thereby refining our understanding of technological themes and providing a more comprehensive, dynamic perspective on their interconnections, ultimately enhancing the accuracy of predictions regarding future technological development trends. Therefore, this study conducts a co-occurrence analysis on the thematic terms extracted above, constructs a thematic term co-occurrence network, and employs three network centrality metrics to filter the thematic terms.

**Word2Vec model.** In the realm of open-source tools for training word embeddings, Word2Vec stands out for its ability to transform unstructured textual information into a vectorized format. By leveraging word vectors, it enhances the contextual connections within texts, thereby providing a more nuanced representation of semantic features. This transformation facilitates more accurate computations of semantic similarity. Fundamentally, Word2Vec constitutes a three-layer neural network architecture, comprising an 'input layer – hidden layer – output layer' configuration [39]. Let us denote the target word as $W(t)$, with its contextual words being $W(t-r)$, $\cdots$, $W(t-1)$, $W(t+1)$, $\cdots$, $W(t+r)$, where n represents the context window size. The framework encompasses two unsupervised learning paradigms: the Continuous Bag of Words (CBOW) model and the Skip-gram model. The CBOW model infers the target word based on its surrounding context, whereas the Skip-gram model reverses this process, predicting the context words given the target word. Illustrations of these models are depicted in Fig 2. In this study, we primarily employ the Skip-gram model to construct a Word2Vec semantic vector space for the topic terms selected earlier. This approach enables a robust representation of the semantic nuances associated with the selected terms, leveraging the strengths of the Skip-gram model in capturing contextual relationships.

**Calculation of topic similarity.** The similarity between topic words from adjacent time series stages is calculated to determine the evolution path between technical topic words. In This study, cosine similarity is used as the calculation metric, with the text vectors calculated above serving as input values. The similarity between two topic words is

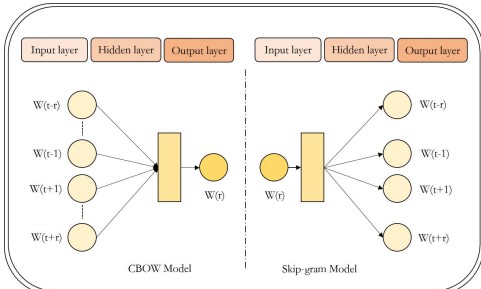

**Fig 2. Network structure of the two Word2vec models.**

determined by calculating the cosine value of the angle between their vectors in adjacent time series stages, as shown in equation (3).

$$Sim(x, y) = \cos \theta = \frac{\sum_i^n x_i y_i}{\sqrt{\sum_i^n x_i^2} \sqrt{\sum_i^n y_i^2}}$$

(3)

$Sim(x, y)$ represents the cosine similarity of vector $(x, y)$; $\cos \theta$ is the cosine value of the angle between vector $(x, y)$, with a range of $[-1, 1]$, a larger cosine value indicates a smaller angle between the vectors, which means higher similarity between the two vectors and a more consistent technical development direction; $i, n$ represent the dimensions of the vectors.

## Evolution path of technical topics

In previous studies, the evolution of technological themes has typically been represented through simple line charts or bar graphs. While these methods can reveal certain trends, they struggle to comprehensively illustrate the interactive changes and migration pathways among themes. In contrast, the Sankey diagram, as a type of flowchart, is commonly used to visualize the structure of information flows in various activities. By employing flow lines with adjustable widths to denote the relationships and intensities between different processes, it effectively showcases the relative importance of these processes. This approach not only clearly presents the strength of relationships between themes but also depicts complex patterns such as the continuation, branching, or merging of themes over time. Consequently, applying the Sankey diagram to multi-source data integration analysis enables researchers to trace the developmental trajectories of themes across different periods within specific technological domains, while also identifying key technological turning points and development trends, thereby significantly enhancing the interpretability and practicality of the analytical results.

In summary, this study visualizes the analysis of technological theme evolution pathways by associating thematic terms from different temporal stages and utilizing Sankey diagrams, thereby clarifying the knowledge flow and accumulation in technology across various time periods.

## Empirical research

### Data acquisition

Through literature review and reference to the retrieval strategy in [40], the data retrieval formulas for This study were formulated, as shown below: ① For CNKI (China National Knowledge Infrastructure) paper retrieval: the subject is (Artificial Intelligence + Computer Vision + Machine Learning + Pattern Recognition + Natural Language Processing + AI + Artificial Intelligence) * Source Category = (CSSCI + CSCD), and the documents must belong to funded literature; ② For OnePatent patent retrieval: TI = (Artificial Intelligence OR Computer Vision OR Machine Learning OR Pattern Recognition OR Natural Language Processing OR AI OR Artificial Intelligence) OR AB = (Artificial Intelligence OR Computer Vision OR Machine Learning OR Pattern Recognition OR Natural Language Processing OR AI OR Artificial Intelligence) AND Patent Status = (Valid) AND Latest Legal Status = (Granted) AND Country = (China); ③ For Wanfang Data Sci-Fund platform fund project retrieval: Project Title: (Artificial Intelligence + Computer Vision + Machine Learning + Pattern Recognition + Natural Language Processing + AI + Artificial Intelligence) * Funding Country: ("China"). Considering the long application period of patents and the lag in public information disclosure, the retrieval time range for all literature data was set from January 1, 2011, to December 31, 2020, with the retrieval date being March 14, 2024. After manual screening to remove irrelevant documents lacking abstract fields, a total of 15,200 papers, 17,858 simple patent families, and 3,515 fund projects were obtained.

## Multi-source data temporal division

In conducting temporal segmentation of multi-source data, this study did not employ linear models or other complex nonlinear models. The rationale lies in the fact that these models often struggle to accurately capture the nonlinear and phased characteristics inherent in the development of artificial intelligence (AI) technologies. Specifically, linear models assume a constant rate of technological advancement, a premise that overlooks the slow accumulation phase in the early stages of technological development, the rapid growth phase in the intermediate stage, and the potential saturation or decline phase in the later stage—a pattern clearly incongruent with the actual development trajectory of the AI field, characterized by rapid iterations and breakthroughs. For instance, in the evolution of AI, breakthroughs in pivotal technologies such as deep learning algorithms have triggered explosive growth in related research and applications within short periods, a phenomenon that linear models fail to explain. On the other hand, while complex nonlinear models can account for a greater number of variables and parameters to simulate the complexity of technological development, they also heighten the risk of model overfitting, thereby diminishing their explanatory power and predictive accuracy in practical applications. Overfitting implies that the model may excessively conform to the noise and intricacies within the training data, resulting in a loss of generalization capability for new data, which is detrimental to research requiring long-term trend analysis and identification of technological development stages.

Taking a holistic view, this study adopts the technology lifecycle S-curve theory, which not only more accurately reflects the dynamic evolution process of AI technology development but also effectively identifies the specific position of technology within its lifecycle, providing a scientific basis for understanding and forecasting the development trends of AI technologies. Consequently, based on this theory, time-stage analysis was performed on the cumulative volume of papers, patents, and funding (corresponding to the sum of papers, patents, and funding in each respective year) using an exponential function, spanning the timeframe from 2011 to 2020. As shown in Fig 3, the output data of papers, patents, and funds have all experienced an introduction period, a growth period, and a maturity period, with $R^2$ values all above 0.95, indicating a good fitting effect. Referring to the peak value K division of the technology lifecycle S-curve in references [41], and considering the publication trend characteristics of the artificial intelligence research field, This study defines K as the maximum slope value of the S-curve fitting, which is the maximum slope value of the exponential fitting curve of the cumulative amount of literature. The range of 0~10% of the curve's peak is the technology introduction period, 10~50% is the technology growth period, and above 50% is the rapid technology growth period. Since artificial intelligence is still a hot topic in current science and technology, This study does not divide the technology decline period in this field. The specific time phase division is as follows:

T1: Technology Introduction Period (2011–2014), during this time phase, artificial intelligence technology, as an emerging field, is in the process of slow development. The number of published papers is nearly 900 per year, most of which are about theoretical introductions and conceptual analyses of the technology in this field, with a relatively small number of patents and funds. Meanwhile, in 2012, AlexNet achieved groundbreaking results in the ImageNet Challenge, attracting exploration and research into deep learning. Therefore, the focus of the artificial intelligence field during this stage is on theoretical and basic research, with significant room for improvement in practical applications.

T2: Technology Growth Period (2015–2018), during this stage, the state issued relevant technical support policies, and artificial intelligence technology experienced rapid development, entering a period of vigorous development of technologies such as deep learning and neural networks. With the advancement of hardware product technology, algorithm optimization, and computational performance improvements, fields such as image recognition, natural language processing, and voice recognition achieved abundant results. People's understanding and application of artificial intelligence have deepened, and the market for related technical demands has gradually expanded. At the same time, the number of related papers, patents, and funds being published is also rapidly increasing.

T3: Rapid Technology Growth Period (2019–2020), during this period, artificial intelligence technology showed new trends of rapid development. Fields such as autonomous driving, smart transportation, intelligent voice, and medical

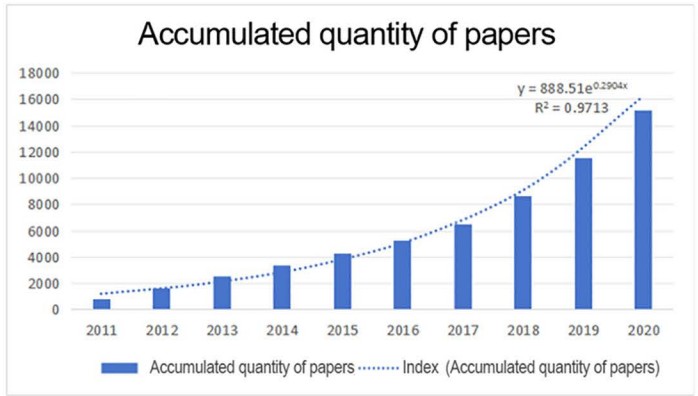

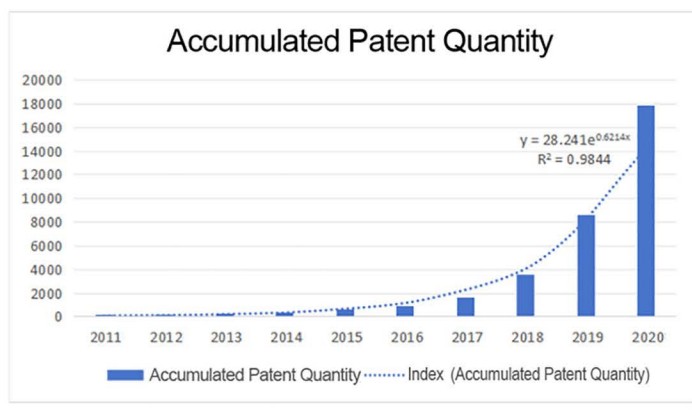

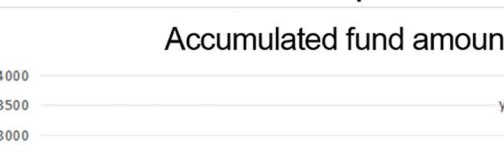

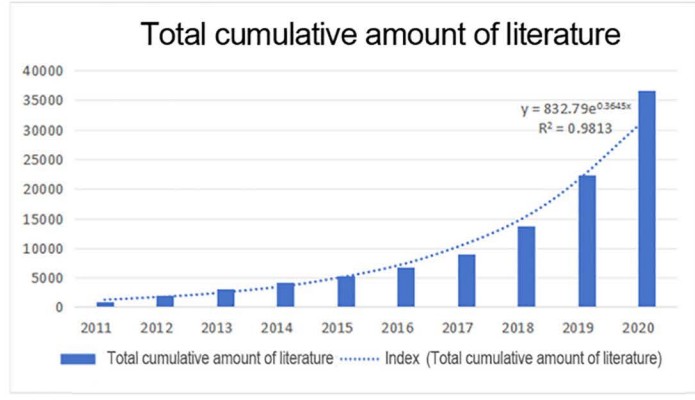

**Fig 3. Comparative Analysis of Multi-source Data Time in the Field of "AI".**

intelligent diagnostics began to emerge. Related application products gradually expanded into consumers' lives, bringing profound changes. The industrialization process of artificial intelligence has accelerated, with more extensive application scenarios, providing new momentum for economic growth. However, as people's understanding of artificial intelligence continues to improve, they have begun to view issues related to ethics, safety, and privacy more rationally, actively participating in discussions and providing suggestions for regulatory strategies. As can be seen from Fig 3, the number of patents during this stage showed a trend of rapid development, growing at a rate of 4000 per year, far surpassing the growth in the number of papers and funds, further illustrating the penetration and development of artificial intelligence technology in practical application scenarios.

### Topic extraction in the field of artificial intelligence

This study employed the jieba toolkit to process abstract data from papers, patents, and funding sources, constructing a specialized vocabulary list for the field of artificial intelligence (AI) to prevent terms such as "artificial intelligence" and "neural network" from being split apart. High-frequency irrelevant words like "the present invention" and "this study" were added to the stop-word list. Using Python, entire Chinese abstracts were segmented with spaces to resemble English-word-like forms, facilitating subsequent topic modeling analysis. During the topic extraction phase, the BERTopic model was chosen over traditional methods such as LDA or TF-IDF, primarily due to its notable advantages in handling complex semantic relationships. Compared to the traditional LDA model, BERTopic excels in capturing the deep semantic

information of words within their contexts, whereas LDA relies solely on word frequency statistics and co-occurrence relationships, failing to effectively identify semantic associations across documents, particularly when dealing with multi-source data. Additionally, LDA is sensitive to parameter settings and susceptible to noise data, leading to less precise topic extraction results. Although TF-IDF can rapidly extract keywords, it lacks an understanding of semantic relationships between words, making it difficult to generate coherent and interpretable topics. In contrast, BERTopic integrates the robust semantic representation capabilities of pre-trained language models (such as BERT), enabling it to effectively identify technological themes and their evolutionary paths in the AI field, while supporting dynamic adjustments to the number of topics, making it suitable for scenarios involving multi-source data integration.

When using the BERTopic model for topic extraction, it is necessary to determine the number of topics in the corresponding field. In This study, the number of topics is set as the result of the model's automatic clustering adjustments. After multiple trials and referring to the parameter settings of related scholars [42], a total of 48 topics are finally obtained and sorted based on the number of documents involved in each topic. As shown in Fig 4, the top 8 topics with the most documents involved are presented, each composed of highly relevant topic words (only the top 5 most relevant topic words are displayed). Topic0 is the topic with the largest number of documents, and the topic words "artificial intelligence," "education," "innovation," "governance," and "economy" mainly summarize the current development of artificial intelligence technology from a macro perspective, relating to politics, economy, education, and other aspects. At the same time, other topics also cover representative technologies such as medical diagnosis, natural language processing, intelligent data analysis, smart transportation, machine learning, robotic equipment, and machine vision.

To further explore the relationships between topics, clustering methods are used to uncover potential clusters among topics, forming a topic distance map. In this map, each circle represents a topic, and the size of the circle corresponds to the number of documents associated with the topic. After iterative similarity calculations of the 48 topics in Fig 5, they can be generally divided into 7 topic clusters, with some exceptions. The largest topic clusters are formed by the following 9 topics: "0_ArtificialIntelligence_Education_Innovation_Governance," "1_Clinical_Prediction_Cell_Model," "16_Sentiment Analysis_Dictionary_Microblog_Features," "21_Chinese_Vietnamese_Natural Language Processing_Character,"

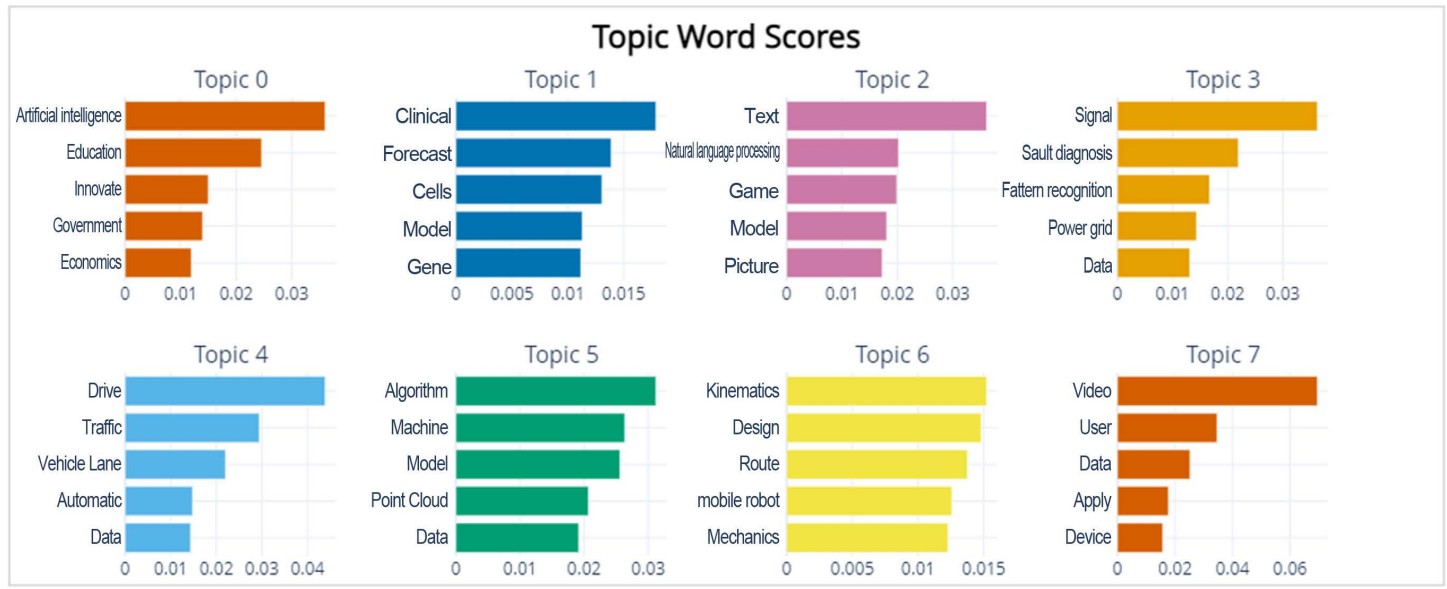

**Fig 4. Score Distribution of Keywords in the Field of "AI" (Partial).**

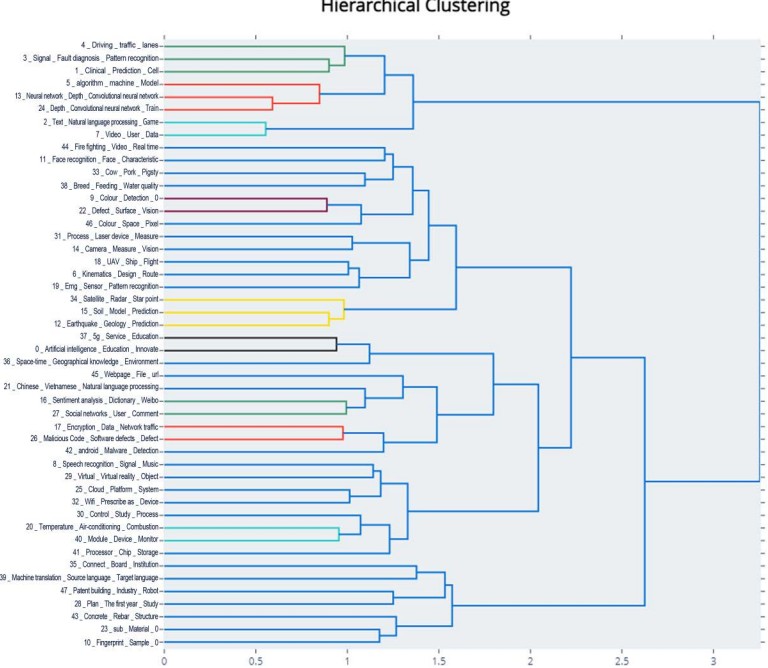

**Fig 5. "AI" Domain Thematic Distance Map.**

"27_Social Network_User_Comments_Rumor," "28_Plan_First Year_Research_Third Year," "36_Spatiotemporal_Geographical Knowledge_Environment_Megalopolis," "37_5G_Service_Education_Network," and "47_Patent Technology_Industry_Robot_Patent Application." This indicates that there is a complex correlation and overlap among the topics within these clusters. The state formulates relevant policies to support artificial intelligence technology, deploys comprehensive strategies for current industrial upgrades, strengthens the construction of digital fields such as smart healthcare, smart cities, and 5G network communications, promotes new forms of productivity, and at the same time, enhances the management of online public opinion. Real-time monitoring of user speech and behavior on network platforms is conducted to reduce the likelihood of rumors.

Hierarchical clustering initially treats each topic in the sample as a separate cluster. Based on the distance between topics, the two topics that are closest to each other are merged into a new cluster, and the distance between the new cluster and other clusters is recalculated. This iterative process is repeated until all topics are merged into a single cluster. Through the hierarchical merging of topics, the final topic hierarchy is presented in the form of a clustering tree, as shown in Fig 6. Segments of the same color represent topics that are merged at the corresponding level.

To deeply analyze the degree of topic association, this study quantifies the relationships between different topics through similarity calculations, creating a 48x48 similarity matrix. A heatmap is used for visualization, as shown in Fig 7. The darker the color of the square, the higher the similarity between the two topics. Overall, the similarity between most topics exceeds 0.5, with the similarity between "0_Artificial Intelligence_Education_Innovation_Governance" and "6_Kinematics_Design_Path_Mobile Robot," as well as between "14_Camera_Measurement_Vision_Stereo" and "25_Cloud_Platform_System_Server" reaching above 0.9.

This study, combining Figs 5–7, further integrates the topics and their corresponding keywords through literature review and manual analysis, as specifically displayed in Table 1. Ultimately, the 48 topics and their vocabulary are consolidated into six primary themes and 23 secondary themes. Overall, the topics covered in this study encompass application

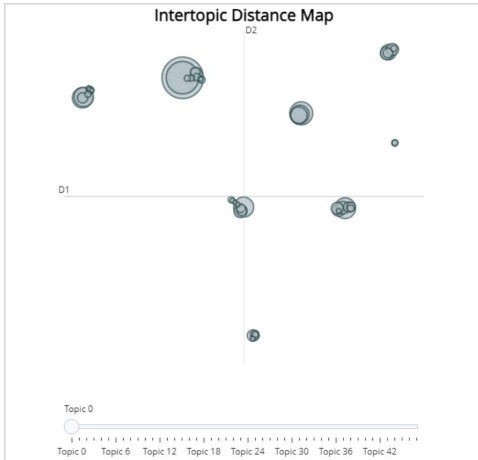

**Fig 6. Hierarchical Clustering Diagram of the "AI" Domain.**

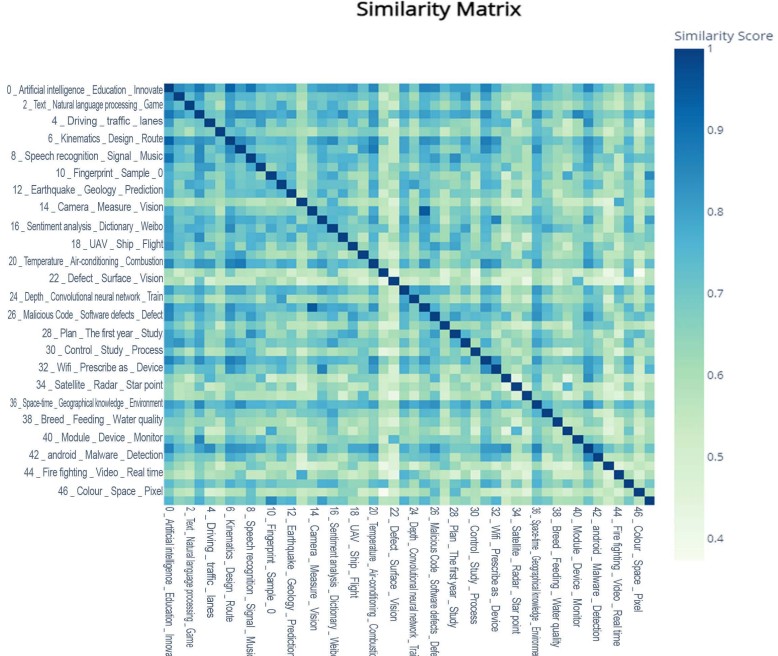

**Fig 7. "AI" Domain Topic Heatmap.**

scenarios of artificial intelligence, network services, applications of intelligent technology, anomaly detection, and human-computer interaction recognition, among others.

## Topic correlation in the field of artificial intelligence

To further explore the accumulation and flow of knowledge among topics, This study refines the research object, uses keywords to represent technological topics, and conducts correlation analysis. First, the top 4 most significant keywords are

**Table 1. Merged topics and subject headings in the field of "artificial intelligence".**

| Primary Theme | Secondary theme | The original theme and vocabulary |
|---|---|---|
| **Theme One: Application and Transformation** | #1Macroeconomic policy support | 0_Artificial Intelligence_Education_Innovation_Governance 28_Plan_First Year_Research_Third Year |
| | #2 Smart City | 4_Driving_Traffic_Lane_Automatic 36_space-time_geography_knowledge_environment_urban agglomeration |
| | #3 Smart Home | 20_Temperature_Air Conditioning_Combustion Data |
| | #4 Smart Healthcare | 1_Clinical_Prediction_Cell_Model |
| **Theme Two: Network Services and Devices** | #5 Cybersecurity | 26_Malicious Code_Software Defect_Defect_Malware 42_Android_Malware_Detection_Application |
| | #6 Privacy protection | 7_Video_User_Data_Application 45_webpage_file_url_browser |
| | #7 Network communication | 17_Encrypted_Data_Network Traffic_Learning 25_Cloud_Platform_System_Server 32_wifi_location_device_terminal 37_5g_ Services_Education_Internet |
| **Theme Three: Applications of Intelligent Technology and Robotics** | #8 Smart equipment and materials | 23_sub_ material_0_polymer 30_Control_Learning_Processing_Motor 35_Connection_Board_Mechanism_Device 41_Processor_Chip_Memory_Module |
| | #9 Biotechnology and Robotics | 6_Kinematics_Design_Path_Mobile Robot 19_Electromyography_Sensor_Mode Recognition_Acceleration 30_Control_Learning_Processing_Motor 47_Patent Technology_Industry Robot_Patent Application |
| **Theme Four: Monitoring and Anomaly Handling** | #10 Astronomy and Aviation | 18_Unmanned Aerial Vehicle_Ship_Flight_System 34_satellites_radar_star points_galaxy |
| | #11 Environmental monitoring | 12_Earthquake_Geological_Prediction_Data 15_Soil_Model_Prediction_5 43_Concrete_Reinforced_Structure_Failure 44_firefighting_video_real-time_forest |
| | #12 Agricultural testing | 10_Fingerprint_Sample_0_Medicinal Materials 33_Cow_Pork_Pig pen _pet 38_Breeding_Feeding_Water Quality_Monitoring |
| | #13 Power Grid Control | 3_Signal_Fault Diagnosis_Pattern Recognition_Grid |
| **Theme Five: Natural Language Processing** | #14 User data | 16_Emotion Analysis_Dictionary_Weibo_Features 27_Social Network_User_Comments_Rumors |
| | #15 Text language | 2_Text_Natural Language Processing_Game_Model 21_ Chinese_Vietnamese_Natural Language Processing_Chinese Characters 39_Machine Translation_Source Language_Target Language_Text |
| **Theme Six: Human-Machine Interaction and Recognition** | #16 Image color detection | 9_Color_Detection_0_Fruit |
| | #17 Visual training | 14_Camera_Measurement_Visual_Binocular 22_Defect_Surface_Visual_Algorithm 24_Deep_Convolutional Neural Network_Training_Visual |
| | #18 Virtual Reality | 29_Virtual_Virtual Reality_Object_Virtual Machine |
| | #19 Scene construction | 5_Algorithm_Machine_Model_Point Cloud |
| | #20 Image scanning | 31_Processing_Laser_Measurement_Scanning 40_Module_Equipment_Monitoring_Sensor |
| | #21 Speech recognition | 8_Voice Recognition_Signal_Music_User |
| | #22 Facial recognition | 11_Face Recognition_Face_Feature_Recognition |
| | #23 Machine learning | 13_Neural Networks_Deep_Convolutional Neural Network_Algorithm |

extracted from each topic, and the keywords from the 48 topics are merged and deduplicated to form a keyword list for the field of artificial intelligence. Meanwhile, the words in each document from different time stages are filtered, removing those not included in the aforementioned keyword list. A co-occurrence network is constructed based on the distribution of keywords in the documents, considering key terms that appear in the same document as forming a co-occurrence connection. By traversing all documents and counting the co-occurrence frequency, the final network effect is shown in Fig 8.

Due to the possibility of the aforementioned keywords repeating multiple times in documents from different time stages, which could result in the same important keywords for each time stage, This study calculates the network's degree centrality, closeness centrality, and betweenness centrality, and performs standard normalization on these measures. By ranking the total values of the three indicators, the top 10 keywords are selected as the fundamental keywords (see Table 2 for specifics). These represent the foundational research knowledge within the field of artificial intelligence and are used as the keywords for the first time stage.

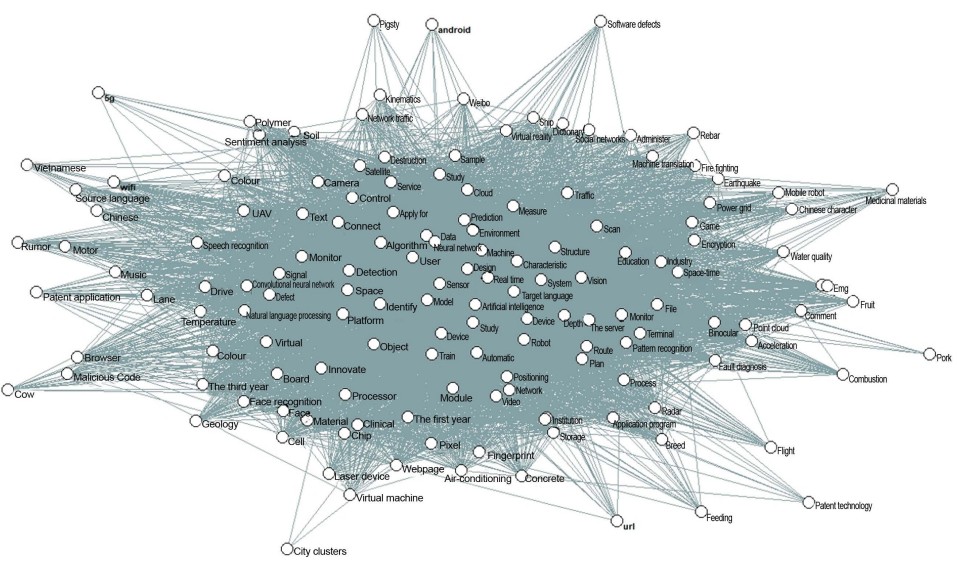

**Fig 8. Co-occurrence Network Diagram of Subject Terms.**

**Table 2. Centrality Indicators of Co-occurrence Network of Keywords (Retaining 3 Decimal Places).**

| Subject term | Degree centrality (Normalized) | Centrality (Normalization) | Centrality of Intermediaries (Normalized) | Summation |
|---|---|---|---|---|
| Learn | 1 | 1 | 1 | 3 |
| Machine | 0.971 | 0.948 | 0.805 | 2.724 |
| Model | 0.971 | 0.948 | 0.778 | 2.697 |
| System | 0.971 | 0.948 | 0.773 | 2.692 |
| AI | 0.943 | 0.898 | 0.755 | 2.596 |
| Data | 0.964 | 0.935 | 0.632 | 2.531 |
| Traits | 0.957 | 0.923 | 0.649 | 2.529 |
| Training | 0.950 | 0.910 | 0.649 | 2.509 |
| Detect | 0.922 | 0.863 | 0.641 | 2.426 |
| Algorithm | 0.922 | 0.863 | 0.597 | 2.382 |

Based on the characteristic of literature knowledge increasing over time, to avoid information overload of subsequent topic evolution, the number of keywords displayed in the T1, T2, and T3 time stages is set to 10, 15, and 20, respectively. By training the Word2vec model, the keywords are represented in vector form, and cosine similarity is used to calculate the topic association degrees between the T1-T2 and T2-T3 stages.

## Verification of result validity

To comprehensively evaluate the advantages of the proposed method over conventional approaches, the authors conducted a systematic comparison of the performance of BERTopic, Word2Vec, TF-IDF, and LDA models in technical topic evolution analysis. The comparison results are identified and presented in Table 3.

As evidenced clearly in the table above, the BERTopic model demonstrates marked superiority over the other three models in terms of Accuracy, Recall rate, and F1 value. Specifically, BERTopic achieves an Accuracy of 76.25%, a Recall rate of 79.16%, and an F1 value as high as 78.12%, indicating its far greater effectiveness in technical topic identification compared to the remaining models. In contrast, while Word2Vec, TF-IDF, and LDA each exhibit distinct performance characteristics, they collectively fall short of BERTopic in overall capability. It is noteworthy that, in addition to its advantages in quantitative metrics, BERTopic also exhibits unique strengths in two critical dimensions: semantic relatedness and dynamic tracking ability. Semantic relatedness refers to the model's capacity to comprehend and capture semantic relationships between documents, which is crucial for precise identification of technical topics. Dynamic tracking ability, on the other hand, denotes the model's proficiency in effectively tracing the evolution of topics within time-series data. These two attributes are of particular importance for interdisciplinary technology foresight and innovative decision-making, as they aid researchers and technological decision-makers in gaining a deeper understanding of the trajectory of technological development and, consequently, formulating more informed strategic plans.

Furthermore, BERTopic's outstanding performance in these areas is primarily attributable to its unique architecture, which is grounded in pre-trained language models (such as BERT). This approach not only enables the model to learn rich contextual information but also allows it to adapt to domain-specific text data through fine-tuning. This provides robust support for processing complex and variable technical literature, making it an ideal tool for technical topic evolution analysis.

In summary, the BERTopic-based method proposed in this paper exhibits exceptional performance in technical topic evolution analysis, particularly in terms of Accuracy, Recall rate, F1 value, semantic relatedness, and dynamic tracking ability, surpassing traditional models such as Word2Vec, TF-IDF, and LDA by a significant margin. This not only validates the effectiveness and superiority of the BERTopic model but also provides reliable methodological support for interdisciplinary technology foresight and innovative decision-making. Future research could further explore the application of BERTopic across a broader range of domains to uncover its potential applications. Additionally, the optimization and refinement of the BERTopic model represent promising avenues for in-depth investigation, aimed at enhancing its performance and expanding its scope of application.

**Table 3. The comparison results of the topic model.**

| Experimental Method | Accuracy/ % | Recall rate/ % | F1 value/ % |
|---|---|---|---|
| **BERTopic (Method in this study)** | 76.25% | 79.16% | 78.12% |
| **Word2Vec** | 72.31% | 77.14% | 75.48% |
| **TF-IDF** | 69.73% | 73.46% | 71.33% |
| **LDA** | 67.29% | 70.64% | 68.79% |

## Topic evolution path in the field of artificial intelligence

This study writes code to convert the topic words and their similarity data divided by time into the format of "source point - endpoint - weight" and saves it in dictionary format. After removing the completely unrelated negative similarity data, the topic word correlation is visualized in the form of Sankey diagram flow lines, as shown specifically in Fig 9. The line width between the square points represents the size of the topic word similarity within adjacent time intervals.

It is clearly evident from Fig 9 that during the evolutionary process from 2011 to 2020 (T1-T2-T3), the similarity of topic words across different time stages exhibits varying degrees of closeness. The association between topic words in the T1-T2 phase is relatively sparse compared to the T2-T3 phase. The correlation degree of topic words such as "detection," "model," "training," and "algorithm" is relatively small. The main reason is that the T1-T2 time phase is a transitional period from the introduction stage to the development stage of artificial intelligence technology, expanding from theoretical research to practical applications. The related technologies and methods are still not mature, resulting in poor correlation between corresponding topic words. However, theoretical topic words such as "artificial intelligence," "learning," and "data" show a stronger correlation.

Within the T2-T3 time phase, artificial intelligence technology rapidly develops and deeply integrates with fields such as biology, medicine, and engineering, giving rise to more interdisciplinary research content. This is particularly evident in the strong correlation within the topic word areas of "natural language processing," "text," "measurement," and "sensors." The overall analysis indicates that the evolutionary analysis path of technological topics in the field of artificial intelligence presents a process from sparse to close correlation between topic words, transitioning from general foundational research to a wide range of application scenarios. At the same time, the optimization and improvement of algorithms such as deep learning and machine learning provide a solid practical feasibility foundation for technological progress and interdisciplinary knowledge integration, promoting the continuous enrichment of knowledge accumulation between topic words and a denser flow of knowledge.

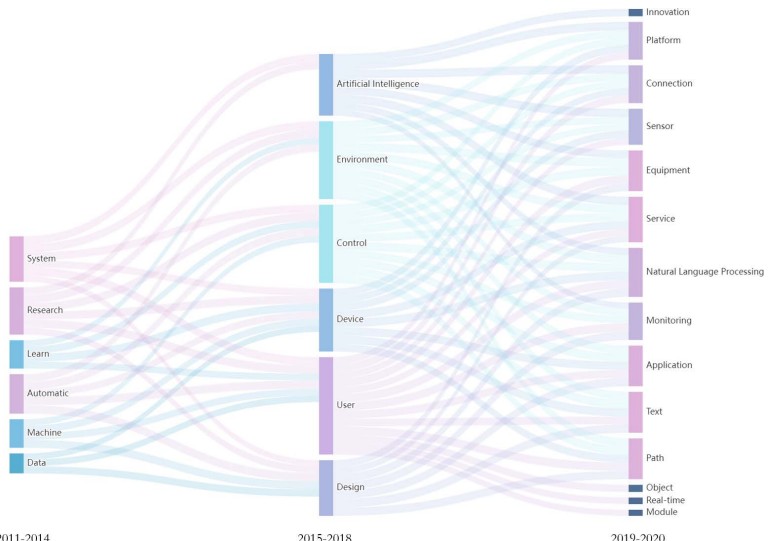

**Fig 9. "Artificial Intelligence" Field Technology Theme Evolution Path Diagram.**

**Interpretation of results**

Based on this, This study analyzes the merged topics in Table 1:

① Topic One: Scenario Application and Transformation. Artificial intelligence technology has gone through a process from preliminary exploration to in-depth application. The "Several Opinions on Promoting Discipline Integration and Accelerating the Training of Postgraduates in Artificial Intelligence in 'Double First-Class' Construction Universities" points out the need to expand technology and innovation methods, achieve the empowerment and transformation of education disciplines through artificial intelligence, and form an "Artificial Intelligence + X" composite development model. At the same time, artificial intelligence technology in smart cities, smart homes, smart medical scenarios, etc., provides convenient services to the public through technologies such as unmanned driving, geographic information monitoring systems, remote control of smart furniture, and automatic diagnostic medical models, which is conducive to accelerating the process of building a digital society.

② Topic Two: Network Services and Equipment. Artificial intelligence technology is widely used in various internet services, optimizing search engines and achieving personalized user recommendations through data mining, machine learning, and other technologies. With the development of 5G network technology, network information services have evolved from early end-to-end to cloud-sharing, making data access more convenient and fast. However, at the same time, security and privacy protection in network transmission have become new issues in the era of big data. The continuous optimization and upgrading of encryption algorithms, and the security detection and timely processing of malicious software and codes, are the keys to solving these issues and are the fundamental guarantees for the long-term stability of a healthy network environment in the future.

③ Topic Three: Smart Technology and Robot Applications. Through artificial intelligence technology, the laws of natural phenomena are simulated and emulated, exploring the optimal path for mobile robots from a kinematic perspective and using sensors to capture electromyographic data for human-computer interaction. The application of industrial robots, service robots, medical robots, and other equipment has significantly improved the efficiency of traditional mechanical work, making the development and design of more intelligent and precise robots one of the important ways to enhance new industrial productivity in the future.

④ Topic Four: Monitoring and Anomaly Handling. The shift from traditional manual monitoring to intelligent monitoring sees artificial intelligence technology playing an irreplaceable role. Satellite surveying, earthquake prediction, drone fire assistance, etc., through real-time monitoring and early warning of data, promptly remind organizational departments to take measures to prevent accidents. In addition, artificial intelligence technology can also be applied in the process of dynamically following up on soil, water quality, and livestock conditions, maximizing the maintenance of excellent conditions.

⑤ Topic Five: Natural Language Processing. Through technologies such as deep learning and natural language understanding, artificial intelligence can perform semantic analysis on large-scale text data, thereby achieving intelligent text processing and language interaction, and by developing intelligent customer service and intelligent assistants, providing users with intelligent language interaction services.

⑥ Topic Six: Human-Computer Interaction and Recognition. To address the poor experience and unmet user needs of traditional interaction methods, smart recognition technology has been introduced into human-computer interaction applications, understanding user behavior. Machine vision, facial recognition, voice recognition, and fingerprint recognition, by identifying attributes such as human actions, appearance, sound waves, and fingerprint features, construct a more intelligent, convenient, and secure interaction method, enhancing user experience satisfaction. In addition, current developments in brain-computer interfaces, virtual reality, and other technologies have become one of the cutting-edge research hotspots for the future.

## Conclusion

### Summary

As an effective method to clarify the future research direction of industrial technology, optimize resource allocation and adjust strategic decision-making, topic evolution analysis plays an important role in improving China 's international competitiveness and becoming a high-quality innovative country. It is particularly important to scientifically and accurately mine technical topics and fine-grained explore the knowledge flow in the evolution process. However, the existing research is limited to a single data source analysis, while ignoring the relevance of text data context. Therefore, this paper collects multi-source data such as related papers, patents, and fund projects in the field of artificial intelligence. Combined with the technology life cycle theory, it divides the technology introduction period, development period, and rapid development period, and extracts abstract data from relevant scientific and technological literature for Bertopic topic modeling. Secondly, through the calculation of three network centrality indicators, 48 technical topics are selected, and the Word-2Vec model is used to calculate the cosine similarity, and the top 10 keywords of the indicators are selected as the basic keywords. Finally, the feasibility of the method is tested by showing the evolution path of the technical theme through the streamline form of the Sankey diagram.

### Contributions

At present, the technology iteration cycle in the field of artificial intelligence is significantly shortened, and the evolution of technology topics presents multi-modal and non-linear characteristics. However, the existing research mostly relies on traditional statistical models such as LDA or a single deep learning method, which is difficult to capture the technical path with strong semantic relevance and dynamic evolution. Therefore, this study first proposes to introduce the BERTopic model into the field of technology topic evolution analysis. Through the deep semantic representation ability of BERT and the c-TF-IDF keyword extraction technology, the problem of insufficient understanding of context semantics by traditional methods is effectively solved. Secondly, the Word2Vec word vector model is innovatively integrated to construct a dynamic word embedding space to realize the temporal semantic drift tracking of technical terms, which makes up for the deficiency of BERTopic in capturing lexical co-occurrence patterns. Finally, a " whole-stage " ambidextrous evolution analysis paradigm is proposed. Combined with the technology life cycle theory, it not only reveals the global evolution trend of technology topics in the field of artificial intelligence, but also accurately locates the combination rules and transition paths of technology hotspots in each stage.It not only provides a methodological innovation with both semantic depth and dynamic adaptability for the research of technology topic evolution, but also expands the analysis framework of multi-model fusion to the interdisciplinary fields of biomedicine, intelligent manufacturing and so on. It provides a scientific basis for policy makers to identify the direction of technological breakthroughs and enterprises to plan R & D paths. At the same time, it lays a methodological foundation for the follow-up research to explore the correlation mechanism between topic evolution and innovation performance.

### Limitations

Due to research capabilities, there are some limitations to this study: ① In addition to the three types of data mentioned above—theses, patents, and fund projects—government documents, online platforms, and social media can also be used as data sources for studying the commercialization of external technology topics, to further explore the economic value of technological topics. ② Based on the analysis of the evolutionary path of technological topics, future research can delve into more refined technological elements, uncovering the behavioral associations between different technological elements.

### Future prospect

In summary, future research can focus on expanding the integration and analysis of multi-source heterogeneous data. By combining multi-dimensional information such as government policy documents, industry white papers, social media

public opinion and market transaction data, this paper reveals the policy-driven mechanism and market response path of technology theme evolution, and provides a more comprehensive empirical basis for the economic value evaluation and innovation diffusion research of technology theme.

Secondly, we should further deepen the research on the relevance of technical elements and deconstruct the underlying elements of technical themes. By constructing a topological network between technical elements, the co-evolution law and competitive exclusion effect are quantitatively analyzed, which provides a micro-level decision-making basis for technology foresight and innovation path selection. At the same time, dynamic network modeling and causal inference algorithms can be developed to deal with high-dimensional and non-linear technical interactions.

Finally, by constructing a three-in-one analysis framework of ' data-model-scene ', a comprehensive research paradigm of dynamic mechanism analysis is built, and the model parameters are optimized in combination with the domain knowledge base, so as to improve the adaptability and prediction accuracy of the method, and promote the research of technology topic evolution to realize the integration and innovation of interdisciplinary methodology.

## Supporting information

**S1 Text.  Text Mining.** https://doi.org/10.6084/m9.figshare.28853663.v1.
(XLSX)

**S2 Text.  The selected subset of topic words.** https://doi.org/10.6084/m9.figshare.28853633.v1.
(TXT)

**S3 Code.  Topic extraction using the BERTopic model.** https://doi.org/10.6084/m9.figshare.28853594.v1.
(PDF)

**S4 Code.  Word2vec model training.** https://doi.org/10.6084/m9.figshare.28853555.v1.
(PDF)

**S5 Code.  Sankey Diagram.** https://doi.org/10.6084/m9.figshare.28853495.v1.
(PDF)

## Author contributions

**Conceptualization:** Xinle Zheng, Kun Lv.

**Data curation:** Xinle Zheng, Qing Wu, Xingyu Luo.

**Formal analysis:** Xinle Zheng.

**Investigation:** Xinle Zheng, Xingyu Luo.

**Methodology:** Xinle Zheng.

**Project administration:** Kun Lv.

**Resources:** Xinle Zheng, Qing Wu, Xingyu Luo.

**Software:** Xinle Zheng.

**Supervision:** Kun Lv.

**Validation:** Xinle Zheng.

**Writing – original draft:** Xinle Zheng, Xingyu Luo.

**Writing – review & editing:** Kun Lv.

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
