## [Decision Letter · Decision Letter 0]

5 Mar 2025

PONE-D-25-08429Analyzing the Evolutionary Trajectory of Technological Themes Based on the BERTopic Model: A Case Study in the Field of Artificial IntelligencePLOS ONE

Dear Dr. Lyu,

Thank you for submitting your manuscript to PLOS ONE. After careful consideration, we feel that it has merit but does not fully meet PLOS ONE’s publication criteria as it currently stands. Therefore, we invite you to submit a revised version of the manuscript that addresses the points raised during the review process.

**Kindly focus on manuscript structure and missing parts.**

A rebuttal letter that responds to each point raised by the academic editor and reviewer(s). You should upload this letter as a separate file labeled 'Response to Reviewers'.A marked-up copy of your manuscript that highlights changes made to the original version. You should upload this as a separate file labeled 'Revised Manuscript with Track Changes'. (highlighted in yellow)An unmarked version of your revised paper without tracked changes. You should upload this as a separate file labeled 'Manuscript'.

We look forward to receiving your revised manuscript.

Kind regards,

Issa Atoum

Academic Editor

PLOS ONE

 [the 2024 Ningbo Soft Science Research Program: Research on the Resource Carrying and Utilization of Ningbo Science and Technology Celebrities - Taking the Revitalization and Development Path of Ningbo's "Hometown of Academicians" as an Example (2024R044);the Zhejiang Soft Science Research Plan Project " Boosting the Development of Zhejiang's Low-Altitude Economy: Current Situation Analysis, Application Scenario Exploration, and Countermeasure Research" (2025C35059)]. 

Additional Editor Comments :

**Kindly spare a separate section for discussion, limitations for better flow even it is available in different parts of the paper.****It is highly recommended to share the data and the code for this project.**

Reviewers' comments:

Reviewer's Responses to Questions

**Comments to the Author**

1. Is the manuscript technically sound, and do the data support the conclusions?

Reviewer #1: Yes

Reviewer #2: Yes

2. Has the statistical analysis been performed appropriately and rigorously? 

Reviewer #1: Yes

Reviewer #2: Yes

3. Have the authors made all data underlying the findings in their manuscript fully available?

Reviewer #1: Yes

Reviewer #2: Yes

4. Is the manuscript presented in an intelligible fashion and written in standard English?

Reviewer #1: Yes

Reviewer #2: Yes

5. Review Comments to the Author

Reviewer #1: The paper presents a novel approach and seems to be interesting. But I recommend a minor revision as I strongly believe the paper can be improve by addressing the following comments:

The figures presented are blurry. It is suggested to use the png or pdfs for better presentations of the graphs in the paper.

The introduction should present motivation, how the authors have arrived at the particular solution and at the end the novel contributions should be presented.

The limitations section has to be presented. It will help researchers to understand the gaps and come up with better research ideas and contribute more to the field.

The future work section is not presented. It will help researchers to analyse the future research directions.

In the discussion section it is recommended to present the comparison of the other algorithms from other papers.

Comparison with baseline models has to be presented in the experimental section. It will help the researchers to understand how well the proposed approach performs in comparison to the other existing and popular models.

Abstract should present main research findings and results of the paper. It will help the readers to find what the research is about.

The introduction section should provide references. Just claims are not enough; authors have to cite the articles supporting the claims.

The uniqueness of the algorithm or proposed approach should be mentioned at the end of the related work section.

Reviewer #2: The paper presents a novel algorithm BERTopic, which is novel. I am suggesting a minor revision as few changes or comments could significantly improve the paper.

Introduction section has to be provided with the references.

The flow between the sections is not consistent. Please maintain the flow from one section to another.

The comparison with the others is not presented. It is recommended to provide the comparison with the other models or frameworks as well.

The abstract does not provide enough details about the research. It is recommended to present the important results in abstract.

The font size of the figures have to be increased. It is not visible properly and blurry.

The contributions are absent. It is suggested to presented contributions in detail.

Comparison with the own models is not presented. It is suggested to present the comparison with the own models as it present the level of rigorous experimentation.

6. PLOS authors have the option to publish the peer review history of their article (what does this mean? ). If published, this will include your full peer review and any attached files.

**Do you want your identity to be public for this peer review?** For information about this choice, including consent withdrawal, please see our Privacy Policy .

Reviewer #1: **Yes: **

Reviewer #2: **Yes: **

---

## [Author Response · Author response to Decision Letter 1]

17 Apr 2025

Response to Reviewer 1 Comments

1. Summary

Thank you very much for taking the time to review this manuscript. Please find the detailed responses below and the corresponding revisions in the re-submitted files.

2. Point-by-point response to Comments and Suggestions for Authors

Comments 1:The figures presented are blurry. It is suggested to use the png or pdfs for better presentations of the graphs in the paper.

Response 1: Thank you for your valuable suggestions regarding the clarity issue of the numbers in the figures. Accordingly, in this round of revision, we have converted the image format to PNG to provide a clearer image quality. We believe that this change will significantly enhance the readability and professionalism of the figures, thereby offering readers a better reading experience.

Comments 2:The introduction should present motivation, how the authors have arrived at the particular solution and at the end the novel contributions should be presented.The introduction section should provide references. Just claims are not enough; authors have to cite the articles supporting the claims.

Response 2: Thank you for your constructive suggestions concerning the comprehensiveness of the content and the structural integrity of the introduction section. In this round of revisions, the author has thoroughly restructured the introduction. Regarding the research motivation, the author has provided an overview of the policy and social backgrounds for the motivation that "the rapid and precise identification of technological development priorities, recognition of technological themes, and elucidation of their evolutionary trajectories through scientific methodologies constitute effective approaches to achieving deep integration between scientific and technological innovation and industrial innovation, laying a solid foundation for future technological strategic planning" Additionally, a substantial number of references have been cited to highlight the novelty of the research perspective presented in this study.

Comments 3:The limitations section has to be presented. It will help researchers to understand the gaps and come up with better research ideas and contribute more to the field.

Response 3:Thank you for your valuable suggestions regarding the completeness of the article. We fully agree with the point about pointing out research limitations to help researchers gain a more comprehensive understanding of research gaps and provide ideas for further studies. At the end of the article, we have specifically added a new paragraph discussing research limitations, which reads as follows: "Due to research capabilities, there are some limitations to this study: ① In addition to the three types of data mentioned above—theses, patents, and fund projects—government documents, online platforms, and social media can also be used as data sources for studying the commercialization of external technology topics, to further explore the economic value of technological topics. ② Based on the analysis of the evolutionary path of technological topics, future research can delve into more refined technological elements, uncovering the behavioral associations between different technological elements."

Comments 4:The future work section is not presented. It will help researchers to analyse the future research directions.

Response 4:Thank you for your invaluable suggestions concerning the completeness of the article. We fully concur with the observation that the article lacks a section on future work prospects, and we recognize that this is a crucial component for guiding subsequent research directions, demonstrating the limitations of the current study, and highlighting potential areas for expansion. At the end of the article, we have specifically added a new paragraph outlining future work prospects. We hope that by providing these prospects, we can inspire more researchers to pay attention to and participate in research in this field, thereby jointly advancing the development of relevant theories and technological progress.

Comments 5:In the discussion section it is recommended to present the comparison of the other algorithms from other papers.Comparison with baseline models has to be presented in the experimental section. It will help the researchers to understand how well the proposed approach performs in comparison to the other existing and popular models.

Response 5:Thank you for the valuable suggestion of adding comparisons with other algorithms and models in the discussion section. We fully agree with this viewpoint and believe that such comparisons not only can more intuitively demonstrate the advantages of the methods we adopted but also provide readers with a comprehensive perspective for understanding. Based on your suggestion, we have added a section on the effectiveness validation of various topic models at line 540 of the article. The experimental results show that the BERTopic topic model exhibits significant superiority in the three key evaluation metrics of accuracy, recall, and F1 score. Additionally, to further enhance the model's performance, we innovatively integrated the Word2Vec word vector model into BERTopic. This approach not only captures semantic information in the text but also effectively leverages word similarities, thereby improving the effectiveness of topic modeling. Through this method, we have not only validated the effectiveness of the BERTopic model but also provided new ideas and methods for future research.

Comments 6:Abstract should present main research findings and results of the paper. It will help the readers to find what the research is about.

Response 6:Thank you for your constructive suggestions regarding the conciseness and completeness of the abstract. In this round of revisions, the author has added the main research contributions and findings of the paper in the corresponding section of the abstract, making the structure of the abstract more complete and better serving as a summary to guide the entire paper.

The paragraph on research findings are as follows: "This study, adopts a multi-source data perspective, employing the life cycle theory to delineate temporal windows. We use the BERTopic model to extract technological themes and construct a co-occurrence network of theme keywords. Three network centrality indices are computed to filter key theme terms, and the Word2Vec model is leveraged to calculate cosine similarities. Ultimately, we map out the evolutionary pathway of technological themes using Sankey diagrams."

The paragraph on research findings is as follows: "Taking the field of artificial intelligence as an example, the study found that the proposed method could effectively identify 48 technical theme keywords and analyze the technological evolution paths of these keywords in areas such as scenario applications, network services, human-computer interaction, intelligent detection, and natural language processing. Furthermore, all evaluation metrics of the model outperformed those of comparable topic models. The rationality of the empirical results was validated through examination against national policies and market application scenarios."

Comments 7:The uniqueness of the algorithm or proposed approach should be mentioned at the end of the related work section.

Response 7:Thank you for your valuable suggestion of mentioning the uniqueness of the algorithm or proposed method at the end of the Related Work section. We have not only added comparisons of algorithm models at line290 in this paper but have also included comparisons with traditional research methods in the literature review, research design, and empirical sections of the article to illustrate the uniqueness of using the BERTopic topic model for analyzing the evolutionary paths of technology topics. For example, at line 154 in the text, we mention: "In contrast, the BERTopic thematic model not only considers the co-occurrence frequency of words but also captures semantic variations of words across different contexts through deep learning techniques, rendering the identification of technological themes more precise and nuanced. Concurrently, the Word2Vec model demonstrates its superiority in excavating the dynamic knowledge flow processes among themes across various timelines. Specifically, by mapping words to a high-dimensional vector space, it generates a context-based distributed representation (i.e., word embeddings) for each word. These word embeddings not only capture semantic relationships between words but also unveil similarities and associations among them through mathematical operations."At the same time, we add the validity test of each topic model in the 540 line of the article. The experimental results show that the BERTopic topic model shows significant advantages in the three key evaluation indicators of accuracy, recall rate and F1 value.

Response to Reviewer 2 Comments

1. Summary

Thank you very much for taking the time to review this manuscript. Please find the detailed responses below and the corresponding revisions in the re-submitted files.

2. Point-by-point response to Comments and Suggestions for Authors

Comments 1:Introduction section has to be provided with the references.

Response 1:Thank you for your valuable comments on our paper, especially regarding the issue of insufficient references in the Introduction section. We are well aware that ample literature citations not only enhance the academic value of the paper but also provide readers with a more comprehensive research background and theoretical foundation. In response to your suggestion, we have carefully reviewed and added several key references in this revision to enrich and improve the content of the Introduction section. We believe that these newly added references will contribute to the completeness and depth of the Introduction section, while also making the entire paper more rigorous and persuasive.

Comments 2:The flow between the sections is not consistent. Please maintain the flow from one section to another.

Response 2:Thank you for pointing out the issue of consistency among different sections of the article. We fully agree and express our sincere gratitude for your meticulous guidance. To enhance the overall coherence and logic of the article, we have carefully examined and adjusted the content arrangement of each section. In the literature review section, we have reorganized it following the sequence of "research on multi-source data - research on technology topics - research on the evolutionary paths of technology topics." This structural adjustment not only makes the literature review more systematic but also helps readers better understand the premises and backgrounds upon which each step of research is based. Additionally, for other sections of the article, we have ensured alignment with the order involved in the research framework diagram. Through these revisions, we hope to significantly strengthen the consistency and logic of the article, enabling readers to follow our line of thought more smoothly and understand the position and role of each section within the entire research framework.

Comments 3:The comparison with the others is not presented. It is recommended to provide the comparison with the other models or frameworks as well.Comparison with the own models is not presented. It is suggested to present the comparison with the own models as it present the level of rigorous experimentation.

Response 3:Thank you for the valuable suggestion of adding comparisons between our model and others. We fully concur with this viewpoint and believe that such comparisons not only present the advantages of our adopted method more intuitively but also provide readers with a comprehensive perspective for understanding. Based on your suggestion, we have added a section on the effectiveness validation of various topic models at line 540 in the article, encompassing not only traditional methods not employed in this paper but also comparisons with the model adopted herein. Experimental results demonstrate that the BERTopic topic model exhibits significant superiority across three key evaluation metrics: accuracy, recall, and F1 score. Additionally, to further enhance model performance, we innovatively integrated the Word2Vec word vector model into BERTopic. This approach not only captures semantic information within the text but also effectively leverages word similarities, thereby improving the efficacy of topic modeling. Through this method, we have not only validated the effectiveness of the BERTopic model but also provided novel ideas and approaches for future research.

Comments 4:The abstract does not provide enough details about the research. It is recommended to present the important results in abstract.

Response 4:Thank you for pointing out that the Abstract section lacks sufficient details about the research and suggesting that key results be presented in the form of an abstract. We fully agree. Therefore, in this round of revisions, the authors have added the main research contributions and outcomes of this paper to make the structure of the Abstract more complete and better serve as a summary to guide the entire text.

The added details are as follows : "Taking the field of artificial intelligence as an example, the study found that the proposed method could effectively identify 48 technical theme keywords and analyze the technological evolution paths of these keywords in areas such as scenario applications, network services, human-computer interaction, intelligent detection, and natural language processing. Furthermore, all evaluation metrics of the model outperformed those of comparable topic models. The rationality of the empirical results was validated through examination against national policies and market application scenarios."

Comments 5:The font size of the figures have to be increased. It is not visible properly and blurry.

Response 5:Thank you for the suggestion that the font size of numbers in the images should be clearer. In this round of revisions, the author has changed the image format to PNG and adjusted the font size of the numbers in the images. We believe this change will significantly enhance the readability and professionalism of the charts, providing readers with a better reading experience.

Comments 6:The contributions are absent. It is suggested to presented contributions in detail.

Response 6:Thank you for your valuable suggestion to add a section on contributions. The author has made references to the contributions in the Abstract, Discussion, and Conclusion sections of the article and has also separately stated the main contributions of this paper at the end of the article, as follows: "At present, the technology iteration cycle in the field of artificial intelligence is significantly shortened, and the evolution of technology topics presents multi-modal and non-linear characteristics. However, the existing research mostly relies on traditional statistical models such as LDA or a single deep learning method, which is difficult to capture the technical path with strong semantic relevance and dynamic evolution. Therefore, this study first proposes to introduce the BERTopic model into the field of technology topic evolution analysis. Through the deep semantic representation ability of BERT and the c-TF-IDF keyword extraction technology, the problem of insufficient understanding of context semantics by traditional methods is effectively solved. Secondly, the Word2Vec word vector model is innovatively integrated to construct a dynamic word embedding space to realize the temporal semantic drift tracking of technical terms, which makes up for the deficiency of BERTopic in capturing lexical co-occurrence patterns. Finally, a " whole-stage " ambidextrous evolution analysis paradigm is proposed. Combined with the technology life cycle theory, it not only reveals the global evolution trend of technology topics in the field of artificial intelligence, but also accurately locates the combination rules and transition paths of technology hotspots in each stage.It not only provides a methodological innovation with both semantic depth and dynamic adaptability for the research of

---

## [Editor Report · Decision Letter 1]

25 Apr 2025

PONE-D-25-08429R1Analyzing the Evolutionary Trajectory of Technological Themes Based on the BERTopic Model: A Case Study in the Field of Artificial IntelligencePLOS ONE

Dear Dr. Lyu,

Thank you for submitting your manuscript to PLOS ONE. After careful consideration, we feel that it has merit but does not fully meet PLOS ONE’s publication criteria as it currently stands. Therefore, we invite you to submit a revised version of the manuscript that addresses the points raised during the review process. **Please ensure that you address all previous comments from the academic editor, particularly those concerning reproducibility. Additionally, kindly revisit all comments and clearly outline the actions taken in the manuscript**

We look forward to receiving your revised manuscript.

Kind regards,

Issa Atoum

Academic Editor

PLOS ONE
---

## [Author Response · Author response to Decision Letter 2]

30 Apr 2025

We take this revision work very seriously and have conducted a thorough re-examination of all modified sections, particularly those concerning reproducibility. Following this round of detailed review and correction, we firmly believe that the version now submitted has reached its optimal state. We appreciate your patience and valuable feedback, which have given us the opportunity to further enhance the quality of our work.

The following is a detailed reply to the comments:

Response to Reviewer 1 Comments

1. Summary

Thank you very much for taking the time to review this manuscript. Please find the detailed responses below and the corresponding revisions in the re-submitted files.

2. Point-by-point response to Comments and Suggestions for Authors

Comments 1:The figures presented are blurry. It is suggested to use the png or pdfs for better presentations of the graphs in the paper.

Response 1: Thank you for your valuable suggestions regarding the clarity issue of the numbers in the figures. Accordingly, in this round of revision, we have converted the image format to PNG to provide a clearer image quality. We believe that this change will significantly enhance the readability and professionalism of the figures, thereby offering readers a better reading experience.

Comments 2:The introduction should present motivation, how the authors have arrived at the particular solution and at the end the novel contributions should be presented.The introduction section should provide references. Just claims are not enough; authors have to cite the articles supporting the claims.

Response 2: Thank you for your constructive suggestions concerning the comprehensiveness of the content and the structural integrity of the introduction section. In this round of revisions, the author has thoroughly restructured the introduction. Regarding the research motivation, the author has provided an overview of the policy and social backgrounds for the motivation that "the rapid and precise identification of technological development priorities, recognition of technological themes, and elucidation of their evolutionary trajectories through scientific methodologies constitute effective approaches to achieving deep integration between scientific and technological innovation and industrial innovation, laying a solid foundation for future technological strategic planning" Additionally, a substantial number of references have been cited to highlight the novelty of the research perspective presented in this study.

Comments 3:The limitations section has to be presented. It will help researchers to understand the gaps and come up with better research ideas and contribute more to the field.

Response 3:Thank you for your valuable suggestions regarding the completeness of the article. We fully agree with the point about pointing out research limitations to help researchers gain a more comprehensive understanding of research gaps and provide ideas for further studies. At the end of the article, we have specifically added a new paragraph discussing research limitations, which reads as follows: "Due to research capabilities, there are some limitations to this study: ① In addition to the three types of data mentioned above—theses, patents, and fund projects—government documents, online platforms, and social media can also be used as data sources for studying the commercialization of external technology topics, to further explore the economic value of technological topics. ② Based on the analysis of the evolutionary path of technological topics, future research can delve into more refined technological elements, uncovering the behavioral associations between different technological elements."

Comments 4:The future work section is not presented. It will help researchers to analyse the future research directions.

Response 4:Thank you for your invaluable suggestions concerning the completeness of the article. We fully concur with the observation that the article lacks a section on future work prospects, and we recognize that this is a crucial component for guiding subsequent research directions, demonstrating the limitations of the current study, and highlighting potential areas for expansion. At the end of the article, we have specifically added a new paragraph outlining future work prospects. We hope that by providing these prospects, we can inspire more researchers to pay attention to and participate in research in this field, thereby jointly advancing the development of relevant theories and technological progress.

Comments 5:In the discussion section it is recommended to present the comparison of the other algorithms from other papers.Comparison with baseline models has to be presented in the experimental section. It will help the researchers to understand how well the proposed approach performs in comparison to the other existing and popular models.

Response 5:Thank you for the valuable suggestion of adding comparisons with other algorithms and models in the discussion section. We fully agree with this viewpoint and believe that such comparisons not only can more intuitively demonstrate the advantages of the methods we adopted but also provide readers with a comprehensive perspective for understanding. Based on your suggestion, we have added a section on the effectiveness validation of various topic models at line 540 of the article. The experimental results show that the BERTopic topic model exhibits significant superiority in the three key evaluation metrics of accuracy, recall, and F1 score. Additionally, to further enhance the model's performance, we innovatively integrated the Word2Vec word vector model into BERTopic. This approach not only captures semantic information in the text but also effectively leverages word similarities, thereby improving the effectiveness of topic modeling. Through this method, we have not only validated the effectiveness of the BERTopic model but also provided new ideas and methods for future research.

Comments 6:Abstract should present main research findings and results of the paper. It will help the readers to find what the research is about.

Response 6:Thank you for your constructive suggestions regarding the conciseness and completeness of the abstract. In this round of revisions, the author has added the main research contributions and findings of the paper in the corresponding section of the abstract, making the structure of the abstract more complete and better serving as a summary to guide the entire paper.

The paragraph on research findings are as follows: "This study, adopts a multi-source data perspective, employing the life cycle theory to delineate temporal windows. We use the BERTopic model to extract technological themes and construct a co-occurrence network of theme keywords. Three network centrality indices are computed to filter key theme terms, and the Word2Vec model is leveraged to calculate cosine similarities. Ultimately, we map out the evolutionary pathway of technological themes using Sankey diagrams."

The paragraph on research findings is as follows: "Taking the field of artificial intelligence as an example, the study found that the proposed method could effectively identify 48 technical theme keywords and analyze the technological evolution paths of these keywords in areas such as scenario applications, network services, human-computer interaction, intelligent detection, and natural language processing. Furthermore, all evaluation metrics of the model outperformed those of comparable topic models. The rationality of the empirical results was validated through examination against national policies and market application scenarios."

Comments 7:The uniqueness of the algorithm or proposed approach should be mentioned at the end of the related work section.

Response 7:Thank you for your valuable suggestion of mentioning the uniqueness of the algorithm or proposed method at the end of the Related Work section. We have not only added comparisons of algorithm models at line290 in this paper but have also included comparisons with traditional research methods in the literature review, research design, and empirical sections of the article to illustrate the uniqueness of using the BERTopic topic model for analyzing the evolutionary paths of technology topics. For example, at line 154 in the text, we mention: "In contrast, the BERTopic thematic model not only considers the co-occurrence frequency of words but also captures semantic variations of words across different contexts through deep learning techniques, rendering the identification of technological themes more precise and nuanced. Concurrently, the Word2Vec model demonstrates its superiority in excavating the dynamic knowledge flow processes among themes across various timelines. Specifically, by mapping words to a high-dimensional vector space, it generates a context-based distributed representation (i.e., word embeddings) for each word. These word embeddings not only capture semantic relationships between words but also unveil similarities and associations among them through mathematical operations."At the same time, we add the validity test of each topic model in the 540 line of the article. The experimental results show that the BERTopic topic model shows significant advantages in the three key evaluation indicators of accuracy, recall rate and F1 value.

Response to Reviewer 2 Comments

1. Summary

Thank you very much for taking the time to review this manuscript. Please find the detailed responses below and the corresponding revisions in the re-submitted files.

2. Point-by-point response to Comments and Suggestions for Authors

Comments 1:Introduction section has to be provided with the references.

Response 1:Thank you for your valuable comments on our paper, especially regarding the issue of insufficient references in the Introduction section. We are well aware that ample literature citations not only enhance the academic value of the paper but also provide readers with a more comprehensive research background and theoretical foundation. In response to your suggestion, we have carefully reviewed and added several key references in this revision to enrich and improve the content of the Introduction section. We believe that these newly added references will contribute to the completeness and depth of the Introduction section, while also making the entire paper more rigorous and persuasive.

Comments 2:The flow between the sections is not consistent. Please maintain the flow from one section to another.

Response 2:Thank you for pointing out the issue of consistency among different sections of the article. We fully agree and express our sincere gratitude for your meticulous guidance. To enhance the overall coherence and logic of the article, we have carefully examined and adjusted the content arrangement of each section. In the literature review section, we have reorganized it following the sequence of "research on multi-source data - research on technology topics - research on the evolutionary paths of technology topics." This structural adjustment not only makes the literature review more systematic but also helps readers better understand the premises and backgrounds upon which each step of research is based. Additionally, for other sections of the article, we have ensured alignment with the order involved in the research framework diagram. Through these revisions, we hope to significantly strengthen the consistency and logic of the article, enabling readers to follow our line of thought more smoothly and understand the position and role of each section within the entire research framework.

Comments 3:The comparison with the others is not presented. It is recommended to provide the comparison with the other models or frameworks as well.Comparison with the own models is not presented. It is suggested to present the comparison with the own models as it present the level of rigorous experimentation.

Response 3:Thank you for the valuable suggestion of adding comparisons between our model and others. We fully concur with this viewpoint and believe that such comparisons not only present the advantages of our adopted method more intuitively but also provide readers with a comprehensive perspective for understanding. Based on your suggestion, we have added a section on the effectiveness validation of various topic models at line 540 in the article, encompassing not only traditional methods not employed in this paper but also comparisons with the model adopted herein. Experimental results demonstrate that the BERTopic topic model exhibits significant superiority across three key evaluation metrics: accuracy, recall, and F1 score. Additionally, to further enhance model performance, we innovatively integrated the Word2Vec word vector model into BERTopic. This approach not only captures semantic information within the text but also effectively leverages word similarities, thereby improving the efficacy of topic modeling. Through this method, we have not only validated the effectiveness of the BERTopic model but also provided novel ideas and approaches for future research.

Comments 4:The abstract does not provide enough details about the research. It is recommended to present the important results in abstract.

Response 4:Thank you for pointing out that the Abstract section lacks sufficient details about the research and suggesting that key results be presented in the form of an abstract. We fully agree. Therefore, in this round of revisions, the authors have added the main research contributions and outcomes of this paper to make the structure of the Abstract more complete and better serve as a summary to guide the entire text.

The added details are as follows : "Taking the field of artificial intelligence as an example, the study found that the proposed method could effectively identify 48 technical theme keywords and analyze the technological evolution paths of these keywords in areas such as scenario applications, network services, human-computer interaction, intelligent detection, and natural language processing. Furthermore, all evaluation metrics of the model outperformed those of comparable topic models. The rationality of the empirical results was validated through examination against national policies and market application scenarios."

Comments 5:The font size of the figures have to be increased. It is not visible properly and blurry.

Response 5:Thank you for the suggestion that the font size of numbers in the images should be clearer. In this round of revisions, the author has changed the image format to PNG and adjusted the font size of the numbers in the images. We believe this change will significantly enhance the readability and professionalism of the charts, providing readers with a better reading experience.

Comments 6:The contributions are absent. It is suggested to presented contributions in detail.

Response 6:Thank you for your valuable suggestion to add a section on contributions. The author has made references to the contributions in the Abstract, Discussion, and Conclusion sections of the article and has also separately stated the main contributions of this paper at the end of the article, as follows: "At present, the technology iteration cycle in the field of artificial intelligence is significantly shortened, and the evolution of technology topics presents multi-modal and non-linear characteristics. However, the existing research mostly relies on traditional statistical models such as LDA or a single deep learning method, which is difficult to capture the technical path with strong semantic relevance and dynamic evolution. Therefore, this study first proposes to introduce the BERTopic model into the field of technology topic evolution analysis. Through the deep semantic representation ability of BERT and the c-TF-IDF keyword extraction technology, the problem of insufficient understanding of context semantics by traditional methods is effectively solved. Secondly, the Word2Vec word vector model is innovatively integrated to construct a dynamic word embedding space to realize the temporal semantic drift tracking of technical terms, which makes up for the deficiency of BERTopic in capturing lexical co-occur

---

## [Editor Report · Decision Letter 2]

4 May 2025

Analyzing the Evolutionary Trajectory of Technological Themes Based on the BERTopic Model: A Case Study in the Field of Artificial Intelligence

PONE-D-25-08429R2

Dear Dr. Lyu,

We’re pleased to inform you that your manuscript has been judged scientifically suitable for publication and will be formally accepted for publication once it meets all outstanding technical requirements.

**I am giving conditional acceptance for the manuscript consideration for PLOS ONE readership, provided these issues are addressed satisfactorily.**

1)   Following the standard structure of academic papers is highly recommended. Kindly rename "Literature Review and Commentary" to "Research Gap" or a related term. Typically, the literature review's final section identifies existing research gaps and sets the stage for the current study. Additionally, the " Summary " subsection under the conclusion might not be necessary, so kindly consider dropping it. Moreover, the paper outline at the end of the introduction is missing; please include it to provide a clear roadmap for the readers.

2)  The sections "Interpretation of Results," "Verification of Result Validity," and "Limitations" could be merged into a separate discussion section. As you have merged results with discussion and numbering, which is not shown, especially with the current paper formatting, please consider the above to ensure better readability.

3)  Ensure consistent spacing between lines throughout the paper. For example, lines 495-512 appear to have double or 1.5-inch spacing. Fix the spacing between words in lines 462-476. Moreover, some text was found in the review response but was not available in the manuscript ("This study adopts a multi-source data perspective").

Kind regards,

Issa Atoum

Academic Editor

PLOS ONE
---

## [Editor Report · Acceptance letter]

PONE-D-25-08429R2

PLOS ONE

Dear Dr. Lyu,

I'm pleased to inform you that your manuscript has been deemed suitable for publication in PLOS ONE. Congratulations! Your manuscript is now being handed over to our production team.

Kind regards,

on behalf of

Dr. Issa Atoum

Academic Editor

PLOS ONE